# Small Heat Shock Proteins in Cancers: Functions and Therapeutic Potential for Cancer Therapy

**DOI:** 10.3390/ijms21186611

**Published:** 2020-09-10

**Authors:** Jixian Xiong, Yuting Li, Xiangyu Tan, Li Fu

**Affiliations:** Department of Pharmacology and Shenzhen University International Cancer Center, Shenzhen University School of Medicine, Shenzhen 518039, China; liyuting@szu.edu.cn (Y.L.); tanxiangyu@szu.edu.cn (X.T.)

**Keywords:** sHSPs, cancer, cancer stem cells, cancer therapy

## Abstract

Small heat shock proteins (sHSPs) are ubiquitous ATP-independent chaperones that play essential roles in response to cellular stresses and protein homeostasis. Investigations of sHSPs reveal that sHSPs are ubiquitously expressed in numerous types of tumors, and their expression is closely associated with cancer progression. sHSPs have been suggested to control a diverse range of cancer functions, including tumorigenesis, cell growth, apoptosis, metastasis, and chemoresistance, as well as regulation of cancer stem cell properties. Recent advances in the field indicate that some sHSPs have been validated as a powerful target in cancer therapy. In this review, we present and highlight current understanding, recent progress, and future challenges of sHSPs in cancer development and therapy.

## 1. Introduction

Although small heat shock proteins (sHSPs) were recognized as protein chaperones several decades ago, they have received considerably less attention for a long time than other heat shock proteins (HSPs). Small heat shock proteins are a class of the superfamily of HSPs with low molecular weight (12–43 kDa) that are ubiquitously expressed in all forms of life. Besides the molecular chaperone activity to prevent the formation of harmful protein aggregates in response to cellular stress, sHSPs are involved in diverse cellular functions such as protein degradation, stress tolerance, cell movement, cell death, cell differentiation, signal transduction, and cell development. As a consequence, sHSPs have important implications in physiological and pathological conditions. In recent years, growing evidence has shown that sHSPs play important roles in various types of cancer, and some sHSPs such as Hsp27 have been proposed to be therapeutic targets for cancer. Thus, understanding sHSPs’ functions and roles in human cancers and elucidating how their malfunction is linked mechanistically to cancers will help identify potential anticancer drug targets, develop effective chemotherapeutic drugs and other therapeutic strategies, and then improve cancer therapy. In this review, we summarize our current understanding of the role of sHSPs in cancers, as well as discuss anticancer drugs and current therapeutic strategies targeting sHSPs.

## 2. Small Heat Shock Proteins (sHSPs)

Small heat shock proteins are a ubiquitous family of ATP-independent molecular chaperones with low molecular mass. Molecular chaperones are proteins that assist other proteins in folding but are not components of these final structures [1,2]. They are classified by their molecular weight into the following families: Hsp110s, Hsp90s, Hsp70s, Hsp60s, Hsp40s, sHSPs, and CCT (TRiC) [3]. sHSPs exist in all kinds of life and play a key role in cellular stress responses.

Like all sHSPs, the primary structures of human sHSPs are characterized by a conserved, structured α-crystallin domain (ACD) flanked by a highly variable amino-terminal region (NTR) and a short flexible carboxy-terminal region (CTR) [4]. The ACD is thought to be essential for the dimerization and function of sHSPs [5]. The NTR is generally hydrophobic [6] and involved in oligomer formation of sHSPs and the interaction with substrate proteins [7]. The CTR is polar, flexible, and plays a key role in the stability and assembly of oligomers [8]. Under in vitro conditions, human sHSPs are often found to exist in a range of oligomeric states, although a few of them, such as HspB6, are reported to be stable and exist as dimers with chaperone activity [9,10]. Recently, HspB1 was reported to exist as phosphorylated monomers, which are from a progressive dissociation of HspB1 oligomers induced by palytoxin in MCF-7 cells and could play a protective role against palytoxin-induced cell death [11]. Similarly, another study showed that oligomer dissociation required only Ser90 phosphorylation of mammalian HspB1, while activation of thermoprotective activity required the phosphorylation of both Ser90 and Ser15 [12]. Further studies are required for understanding the role of phosphorylation of HspB1 in the interconversion of HspB1 ultrastructures between oligomeric/dimeric/monomeric forms and the regulation of its functions.

The human sHSPs contain 10 members, including HspB1 (Hsp27), HspB2 (myotonic dystrophy kinase-binding protein, MKBP), HspB3 (Hsp17), HspB4 (αA-crystallin), HspB5 (αB-crystallin), HspB6 (Hsp20), HspB7 (cardiovascular Hsp, cvHsp), HspB8 (Hsp22), HspB9 (cancer/testis antigen 51, CT51), and HspB10 (outer dense fiber protein 1, ODFP1). Some (HspB1, HspB5, HspB6, HspB8) are ubiquitously expressed in various tissues, while others are expressed in specific tissues (Table 1). In eukaryotes, the expression of sHSPs is under the control of the heat shock factor (HSF) transcription factors, which can initiate transcription of sHSPs genes and upregulate the expression of sHSPs in response to stress [13,14]. Moreover, sHSPs themselves are sensitive to their conditions, and their expression of proteins is coordinated by cellular conditions [15].

sHSPs are best known as molecular chaperones that bind various non-native (misfolded or unfolded) substrate proteins, prevent their uncontrolled aggregation, and facilitate the refolding of these proteins independently or by the cooperation of ATP-dependent chaperones. Usually, sHSPs do not exhibit refolding activities but capture the unfolded proteins and stabilize them in sHSP/substrate complexes. Refolding and release of bound substrates are under the help of other ATP-dependent chaperones [18,19,20]. In addition, sHSPs are involved in a growing number of other cellular functions, including cell protection, cellular signaling, cell differentiation, cell movement, cell apoptosis, and cell development [21,22]. Whether and how the chaperone and nonchaperone activities of sHSPs are related raise questions for the future. Pathologically, sHSPs have been reported to be linked to a number of diseases, such as cataracts, neurological disorders, myopathies, multiple sclerosis, aging, and cancers [21,22].

## 3. Functions of Small Heat Shock Proteins in Cancers

Although the original focus on the role of sHSPs revealed its chaperone functions, work over several decades has shed light on their important roles in cancers. Hsp27 (HspB1) was initially found to be expressed in meningiomas [23]. Increasing findings indicate that sHSPs are expressed in diverse malignancies and have been linked to several hallmark features of cancer, including tumorigenesis, cell growth, apoptosis, metastasis, and chemoresistance, as well as cancer stem cells.

### 3.1. Expression of Small Heat Shock Proteins in Cancers

Small heat shock proteins are ubiquitously expressed in numerous types of tumors, including head and neck (HspB1, HspB5) [23,24], breast (HspB1, HspB2, HspB5, HspB8) [25,26,27,28,29,30], cervical (HspB1) [31], colonrectal (HspB1, HspB5, HspB6, HspB9) [32,33,34,35], esophageal (HspB9) [35], gastric (HspB1, HspB5, HspB8) [36,37,38], larynx (HspB1, HspB5) [39,40], liver (HspB1, HspB5) [41,42], lung (HspB1, HspB5, HspB9) [35,43,44,45], oral (HspB5) [46], ovarian (HspB1, HspB8) [47,48], pancreatic (HspB4, HspB9) [35,49,50], prostate (HspB1) [51], renal (HspB1, HspB5, HspB7) [52,53,54], testis (HspB9) [35], cancers and glioblastoma (HspB1, HspB5) [55,56], and osteosarcoma (HspB5, HspB8) [57,58].

Increased levels of sHSP expression were identified in breast (HspB1, HspB2, HspB5) [25,26,27,28,29], cervical (HspB1) [31], colorectal (HspB1, HspB5) [32,33], gastric (HspB1, HspB5, HspB8) [36,37,38], glioblastoma (HspB1, HspB5) [55,56], larynx (HspB1, HspB5) [39,40], liver (HspB5) [42], lung (HspB1, HspB5) [43,45], oral (HspB5) [46], osteosarcoma (HspB5) [57], ovarian (HspB8) [48], prostate (HspB1) [51], renal (HspB1, HspB5) [52,53], and testis (HspB9) [35] cancers. Conversely, decreased expression of sHSPs was observed in colorectal (HspB6) [34], pancreatic (HspB4) [49,50], and renal (HspB7) [54] cancers. Additionally, sHSP expression is closely associated with the progression of cancers and poor clinical outcomes. We summarized the levels of sHSP expression in various cancers (Table 2).

### 3.2. Tumorigenesis

Recent studies have indicated that sHSPs are related to the transformation into neoplastic cells. sHSPs can enhance or suppress tumorigenesis and cancer progression. Overexpression of HspB5 in human mammary epithelial cells (MECs) led to morphological, physiological transformation, and carcinomas formation in vivo [28]. Moreover, MECs with HspB5 overexpression showed neoplastic characteristics, including enhancing cell growth, migration and invasion, and inhibiting apoptosis. Conversely, HspB4, HspB6, and HspB7 are reported to suppress tumorigenesis in pancreatic, colorectal, and renal cancer, respectively [34,49,50,54]. In addition, HspB7 expression alteration was reported to be controlled by epigenetic regulation [54]. DNA hypermethylation is inversely correlated with the mRNA level of HspB7, and 5-aza-2′-deoxycytidine (5-Aza-dC) treatment elevated the expression of HspB7 [54]. These data suggested that methylation-dependent expression regulation of HspBs may be important for tumorigenesis of cancer cells. However, the mechanisms governing HspBs expression are not fully understood. Interestingly, HspB4 (αA-crystallin) and HspB5 (αB-crystallin) are two isoforms of α-crystallin, while existing evidence indicates that they may play different roles in tumorigenesis: HspB4 acts as a suppressor, and HspB5 acts as a promoter. Further investigation of the mechanism will clarify the divergent roles of sHSPs in tumorigenesis.

### 3.3. Cell Growth, Death, and Tumor Development

sHSPs have been identified to play a pivotal role in cell proliferation and survival and cancer progression. HspB1 can enhance or inhibit cell proliferation and growth and tumor development. Studies have shown that overexpression of HspB1 promotes cell proliferation and growth of breast cancer cells in vitro [15]. However, HspB1 was also suggested as a suppressor for cell proliferation in human testis tumor cells [59]. HspB5 overexpression has been reported to promote tumor growth of xenografts derived from breast cancer cells [60]. Another sHSP protein, HspB8, has been recently identified to enhance cancer progression. The knockdown of HspB8 expression inhibited in-vitro cell proliferation of breast cancer cells [61], while HspB8 overexpression enhanced cell proliferation and growth of breast cancer [61]. Interestingly, studies have suggested that HspB8 might play an important role in estrogen response and breast cancer progression. HspB8 is regulated by estrogen and can augment cancer cell progression by estrogenic stimuli [62,63,64], suggesting HspB8 might be a target associated with ER-positive tumors. However, the molecular mechanism should be further explored.

sHSPs protect cells from death by preventing the aggregation of denatured proteins and interacting with various components of the death-associated signaling pathways. Growing evidence suggests that most of the sHSPs (HspB1, HspB2, HspB5) possess an antiapoptotic activity and inhibit apoptotic cell death in various malignancies. The expression of HspB1, HspB2, and HspB5 is reported to be associated with resistance to apoptosis in human breast cancer cells [27,60] and oral verrucous carcinoma [65]. Overexpression of HspB1, HspB2, and HspB5 inhibits apoptosis in breast cancer cells [27,60,66]. Moreover, knockdown of HspB1 in-vitro induced apoptotic cell death, whereas HspB1 upregulation reduced apoptosis and enhanced tumorigenic potential in-vivo in glioblastoma multiforme [67]. Exceptionally, HspB6 was recently recognized to induce apoptosis in colorectal cancer [34]. Additionally, HspB8 is suggested to have either pro- or antiapoptotic effects in a context-dependent manner [68,69]. Several mechanisms for sHSPs regulating apoptosis by integrating with different signaling pathways have been suggested, including extrinsic and intrinsic apoptosis (Figure 1): (a) sHSPs (HspB2) inhibit the extrinsic apoptotic pathway by inactivating caspases-8, 10 [27]; (b) sHSPs (HspB1, HspB5) inhibit the activation of caspase-3 to block apoptosis [67,70,71]; (c) sHSPs (HspB1, HspB5) can regulate bax, bak, and other members of the Bcl-2 family or interact with Bcl-2 to sequester its translocation to mitochondria, thereby preventing the activation of the intrinsic apoptotic pathway [72,73]; (d) sHSPs (HspB1) block cytochrome c [74] and Smac [75] released from mitochondria or bind with cytochrome c released from mitochondria [76], then preventing the activation of caspase-9 and caspase-3; (e) sHSPs (HspB1, HspB5) inhibit the p53-dependent activation of the Bcl-2 family members, thus indirectly inhibiting its proapoptotic effect against apoptotic Bcl-2 proteins [77,78]; (f) sHSPs (HspB1) can also interact with protein kinase C (PKC) delta type [79] or nuclear factor of kappa light polypeptide gene enhancer in B-cell inhibitor (IkB), alpha [80], thereby inhibiting caspase-3 activation. It is important to note that the phosphorylation states of sHSPs have been reported to regulate cancer cell death. Early studies have shown that both phosphorylated and unphosphorylated HspB1 inhibit apoptosis [77,78]. Recently, several studies have indicated that phosphorylation of HspB1 increases apoptosis in leukemia cells [70], and phosphorylation of both HspB1 and HspB5 increases apoptosis in breast cancer cells [73,81]. It is given that sHSPs undergo a number of post-translational modifications (PTMs) under physiologic and pathologic conditions (e.g., phosphorylation, acetylation, and glycosylation). In addition, these PTMs alter sHSP structure, then modulate apoptotic activity by interacting with different proteins and playing significant roles in regulating cell survival. However, our understanding of how these PTMs impact sHSPs’ apoptotic function is clearly incomplete. Thus, the significance and roles of sHSPs’ PTMs in cancer cells will need further investigation.

In addition, sHSPs play cytoprotective roles against apoptosis induced by oxidative stress in cancer cells, and their high expression in cancer cells results in cancer cell radio- and chemoresistance. As a byproduct of oxygen metabolism, reactive oxygen species (ROS) induce oxidative stress, damage macromolecules such as proteins, lipids, and nucleic acids, and, therefore, trigger apoptotic pathways and result in cell death. Cells use the antioxidant system (e.g., antioxidant enzymes, GSH, and CoQ as internally synthesized antioxidants, and vitamin E and carotenoids as externally supplied antioxidants) to neutralize free radical cellular damage and maintain normal redox homeostasis. Unlike normal cells, many cancer cells develop efficient antioxidant systems to defend against oxidative damage caused by ROS to prevent cell death. Studies have shown that the expression of HspB1 can significantly decrease cellular the ROS level and upregulate the total glutathione level [82,83]. HspB1 may act in different ways to modulate cell intracellular redox status: (a) decreasing free radical production by reducing the activities of enzymes, including superoxide dismutase (SOD) [84,85] and catalase [85,86], or decreasing of intracellular iron to prevent its participation in free radical formation [87]; (b) scavenging free radicals by modulating the activity of antioxidant enzymes such as glucose 6-phosphate dehydrogenase (G6PDH) [88,89], glutathione reductase (GR) [90], and glutathione peroxidase (GPx) [85,90] to increase reduced glutathione (GSH) levels and downregulate ROS; (c) activating transcription factors, such as nuclear factor erythroid 2-related factor 2 (NRF2), thereby activating their target antioxidative enzymes [91]. Consistent with the abovementioned mechanisms, increased levels of sHSPs, together with multiple antioxidant molecules, including superoxide dismutase (MnSOD), thioredoxin reductase 2 (TXNRD2), glutathione (GSH), glutathione peroxidase (Gpx), were identified in thyroid tumors [92]. In fact, all antioxidants work cooperatively as a complex network to maintain optimal redox balance. The roles of sHSPs in redox regulation in different cancer cells and how sHSPs coordinate the antioxidant defense network to protect against oxidative stress-induced cell death will need further investigation. Furthermore, current reports indicate that HspB1 is associated with therapeutic resistance in colon [93], liver [94], and head and neck [95] cancer cells through the regulation of ROS levels; thus, targeting sHSPs could be a potent strategy to help overcome chemo/radioresistance in cancers.

### 3.4. Cell Migration/Invasion, Angiogenesis, and Tumor Metastasis

Consistent with sHSPs’ enhanced expression in many malignancies, increasing reports have demonstrated that sHSPs play important functional roles in cell migration, invasion, and angiogenesis. Initially, HspB1 was reported to promote the invasion of breast cancer cells [96] by upregulation of MMP-9 expression [97] but decrease the motility of breast cancer cells [96]. Subsequently, numerous studies have demonstrated that overexpression of HspB5 in breast cancer cells increased cell migration and invasion [28], and HspB1 silencing in colorectal [98], prostate [99], ovarian [100], and liver [101] cancers or head and neck squamous cell carcinoma [102], HspB8 silencing in breast cancer [61], or HspB5 silencing in colorectal [103] and renal [53] cancers inhibit migration and/or invasion of cancer cells. Moreover, a number of clinical data studies have shown that primary tumor expression of sHSPs is associated with aggressive tumor characteristics and tumor progression. For example, HspB1 mRNA and protein expression correlate with peritoneal metastasis and poor survival in ovarian cancer [104]. More recently, high serum HspB1 expression in ovarian cancer [105] and renal cancer [106] has been shown to be associated with tumor metastasis. Additionally, phosphorylated HspB1 is also reported to be associated with portal vein invasion in hepatocellular carcinoma [41] and lymph node metastasis in breast cancer [26]. Additionally, HspB5 expression correlates with the tumor–node–metastasis (TNM) stage and predicts poor survival in non-small-cell lung cancer [45]. In breast cancer, HspB5 expression is an independent predictor of brain metastasis and poor survival and can also predict brain metastasis as the first site of distant metastasis [107,108]. Although overwhelming data have indicated that sHsps can promote cell migration and invasion and tumor metastasis, a few notable exceptions have been reported. HspB1 mRNA and protein levels were lower in tumor tissue in a small study of thyroid carcinomas [109]. Additionally, a few studies showed that HspB5 expression was not significantly correlated with prognosis in head and neck carcinomas [110] and lung cancer [111]. Moreover, HspB4 has been reported to suppress pancreatic cancer cell migration [49,50]. Thus, it should be clarified whether these divergent and even contrasting results reflect differences in study design and/or tumor type. Although the mechanisms underlying sHSPs’ effects on cell migration/invasion have not been clearly defined, results have shown that sHSPs promote migration, invasion, and metastasis, likely by regulating MMP expression [97], by interacting with and regulating intermediate filament dynamics [98], or via epithelial–mesenchymal transition (EMT) [103]. Additionally, it is noteworthy that phosphorylation may be important for sHSPs’ effects on these actions [26,39,112], and understanding the roles and functions of sHSPs’ PTMs on cell migration/invasion will deepen our understanding of sHSPs’ roles in malignancy.

Tumor angiogenesis, the process where tumors form new blood vessels from pre-existing ones, plays critical roles in tumor growth and metastasis [113]. HspB1 has been reported to be induced during tumor angiogenesis [114]. HspB5 was identified to promote tumor angiogenesis by modulating tubular morphogenesis and survival of endothelial cells [115]. Similarly, HspB1 has been reported to induce angiogenesis by increasing vascular endothelial growth factor (VEGF) [116]. Additionally, HspB1 expression was suggested to be associated with the expression and secretion of angiogenesis-associated proteins such as VEGF-A [117,118]. Consistent with this finding, HspB5 has been reported to regulate tumor angiogenesis by modulating VEGF-A [119]. Moreover, HspB1 has been identified as the target molecule of angiogenesis inhibitors [120]. Collectively, these data suggest that sHSPs have a proangiogenic effect and might be potential targets for antiangiogenic cancer therapy.

### 3.5. Chemoresistance

sHSPs are a group of molecular chaperones protecting cells by maintaining cellular homeostasis. HspB1’s murine homolog Hsp25 expression is induced significantly by treatment with chemotherapy drugs such as cisplatin and doxorubicin in Ehrlich ascites tumor (EAT) cells [121,122]. Additionally, the upregulation of HspB1 was also reported in cervical cancer cells and prostatic cancer cells by treatment with 17-allylamino-demethoxygeldanamycin (17-AAG) [123]. Moreover, HspB1 was significantly higher in geldanamycin-resistant A549 NSCLC cells [123]. Furthermore, HspB1 exhibited an alteration of its degree of phosphorylation when treatment with vinblastine, paclitaxel, and doxorubicin in breast cancer cells and serine 59 phosphorylation of HspB1 induced apoptosis of vinblastine-treated breast cancer cells [73], suggesting that specifically inducing the phosphorylation of HspB1 can improve therapeutic outcomes by circumventing the drug resistance of breast cancer. Artificial overexpression of HspB1 increased doxorubicin resistance of breast cancer cells [15], cisplatin and doxorubicin resistance of testis tumor cells [59], and 17-AAG resistance of cervical cancer cells [123]. Similarly, knockdown of HspB1 decreased doxorubicin resistance of breast cancer cells [15] and 17-AAG resistance of cervical cancer cells [123]. Collectively, these studies suggest that HspB1 has a significant role in chemotherapy drug resistance. Remarkably, it has been demonstrated that inhibition of Hsp90 is associated with the upregulation of HspB1. Hsp90 forms complexes with HSF1 and maintains HSF1 in a repressed, transcriptionally inactive form. Inhibitors or binding agents that target Hsp90 can disrupt Hsp90-HSF1 interaction, thereby freeing HSF-1 by dissociation from Hsp90. Activated HSF-1 is translocated into the nucleus, initiates transcription of previously silent Hsp genes, including HspB1, Hsp40, and Hsp70, and leads to the activation of a prosurvival process called the heat shock response, thus limiting the activity of Hsp90 inhibitors and contributing to drug resistance and toxicity. Studies show that a combination of downregulation of HspB1 by gene silencing or inhibitors and Hsp90 inhibitors could reduce drug resistance and enhance the efficacy of Hsp90-directed therapy [123,124]. However, other sHSPs’ roles in resistance to anticancer treatment are not clearly defined, and further in-depth investigations need to be followed to expand the roles of sHSPs in cancer therapy.

### 3.6. Cancer Stem Cells

Cancer stem cells (CSCs), also termed “tumor-initiating cells” (TICs) or “sphere-forming cells”, are a small subpopulation of stem-cell-like cancer cells that present the stem cell property of self-renewal, generating differentiated tumor cells and resistance to radiochemotherapy [125]. CSCs have been isolated from numerous solid and liquid tumors, including but not limited to leukemia, brain, head and neck, lung, breast, liver, stomach, colon, pancreatic, prostate, ovary, and bladder cancers and mesothelioma [126]. CSCs are believed to be responsible for cancer pathogenesis, namely, cancer initiation, recurrence, metastasis, and drug resistance, and are involved in the progression of human malignancies [127]. Thus, targeting CSCs and the elimination of CSCs have been considered as an emerging area in cancer therapy.

Current reports indicate that HspB1 plays a crucial role in regulating CSC-associated properties such as survival, stemness, high migration activity and invasiveness, and chemoresistance. Initially, HspB1 was reported to be associated with chemoresistance of lung cancer stem cells [128]. Then, HspB1 was identified to be related to CSC stemness by proteomics [129]. Indeed, several studies have shown that HspB1 is essential for CSC stemness in salivary adenoid cystic carcinoma [130], non-small-cell lung cancer [131], and breast cancer [132]. There are also reports linking HspB1 to the maintenance of CSC phenotypes in breast cancer stem cells [133] and gynecologic cancer stem cells [134]. Additionally, the contribution of HspB1 to cell migration, invasiveness, and EMT was found in several CSC-related studies, including breast cancer [133] and salivary adenoid cystic carcinoma [115]. Similarly, HspB1 was identified to regulate vasculogenic mimicry activity in CD24^−^CD44^+^ALDH^+^ breast cancer stem cells [135]. Furthermore, numerous reports have suggested that HspB1 is key for CSC survival and contributes to radio- and chemoresistance. For example, HspB1 is considered to be essential for the survival of CD133^+^ colon cancer stem cells [136]. Additionally, HspB1 (along with Hsp70) is upregulated in radioresistant CSC-like SP cells isolated from breast cancer cells [137]. In addition, HspB1 resistance to apoptosis, hyperthermia, and chemotherapeutic agents was reported in oral CSCs [138], lung CSCs [128,139], breast CSCs [132,140], and esophageal CSCs [141].

Importantly, it should be noted that HspB1 involvement in modulating CSC properties depends on the levels of its expression and activity. Though HspB1 can be expressed constitutively in mammalian cells, its expression is mainly coordinated by HSF1 (heat shock transcription factor 1), which is a stress-responsive transcriptional factor that induces the transcriptional activation and the expression of HSPs such as HspB1 and Hsp70 [13]. Under any proteotoxic stress (heat, hypoxia, energy starvation) stimuli, HSF1 is activated and translocated into the nucleus, where it activates HSP gene transcription. Additionally, HSF1 activation/inactivation mainly depends on its phosphorylation [142]. Indeed, phosphorylation of HSF1 at serine 326 was reported to be critical for the maintenance of gynecologic CSCs by induction of HspB1 [134]. Besides HspB1 expression regulated by HSF1, the inhibition of HspB1 by SMURF2 (SMAD ubiquitin regulatory factor 2) is reported to be involved in repressing the self-renewal capability of breast CSCs [132], indicating that ubiquitin-dependent protein degradation seems to play a role in modulating HspB1 expression and CSC properties. It is generally accepted that the oligomerization and cytoprotective activities of HspB1 are mainly regulated by phosphorylation. Recently, numerous studies have confirmed that HspB1 phosphorylation/dephosphorylation is a critical regulator in the formation/maintenance of CSC properties. HspB1 undergoes phosphorylation as the terminal substrate in the p38/MAPK pathway, and there are increasing reports linking this p38 MAPK/MAPKAPK2/HSP27 pathway to chemoresistance in lung CSCs [128,131] and oral CSCs [138], maintenance of CSC properties in lung CSCs [131], and EMT in renal [143] and lung CSCs [139]. Similarly, in colorectal CSCs, protein phosphatase PP2A was reported to dephosphorylate HspB1 and attenuate HspB1 effects by promoting CSC properties [136,144], indicating that the status of phosphorylated HspB1, mediated by kinase and phosphatase, is essential for HspB1 effects on CSC properties. However, the detailed regulatory mechanism of HspB1 phosphorylation that depends upon the temporal and spatial balance of kinase and phosphatase should be investigated further. Conversely, inactivation of p38 has been suggested to promote CSC properties in non-small-cell lung cancer cells [145]. In this case, phosphorylated HspB1 by the p38/MK2 pathway can interact with the stemness-related proteins such as SOX2, OCT4, NANOG, KLF4, and c-Myc, then promote their ubiquitination and degradation, suggesting that phosphorylated HspB1 may have different roles in various types of CSCs by interacting with the effector proteins.

Large numbers of publications have demonstrated that HspB1’s maintenance and modulation of CSC characteristic features are largely dependent on its interaction with client proteins. Evidence has shown that HspB1 can inhibit apoptosis by interacting with procaspase-9 and procaspase-3 to prevent activation of caspase-9 and caspase-3 [128]. The direct interaction between HspB1 and AKT in esophageal CSCs is suggested to be critical for the maintenance of CSC features such as a high metabolic rate [141]. Similarly, we report that the cancer stemness-related activities of Hsp90 are regulated by clusterin through Hsp90 direct interactions with its client protein, AKT, in gastric cancer stem cells [146]. In addition, HspB1 regulates the maintenance and EMT of breast CSCs by interacting with IkB, inducing its degradation, and activating NFkB [133]. Consequently, among HspB1’s interacting proteins, there are surely some proteins that contribute to CSC-driven cancer development or are responsible for the manifestation of certain properties of CSCs. However, HspB1’s interacting proteins and their implications and roles in the formation/maintenance of the CSC phenotype remain to be elucidated.

Taken together, these published data indicate that HspB1 is responsible for the maintenance/modulation of CSC properties such as the stemness and self-renewal of CSCs, their ability to promote EMT and metastasis, their resistance to apoptosis and radiochemotherapy, and their energy metabolism switching [147] (Figure 2). No doubt, HspB1 is one of the most promising targets for CSC-based cancer treatment. However, several points should be addressed for better understanding of sHsps’ roles in CSC maintenance and progression. First, the mechanisms leading to HspB1 expression alteration in CSCs are not fully understood. Up until now, HspB1 expression alteration in CSCs has been mainly focused on the transcriptional level; for example, the transcriptional factor HSF1. Other mechanisms, such as microRNA and epigenetic regulation governing HspB1 expression, remain elusive. Second, how do other PTMs’ roles in modulating CSC properties regulate HspB1 activity? Third, current publications are mainly focused on the roles of HspB1 in CSCs. How do other sHsps play a role in CSCs and the underlying mechanisms? Answering these questions will deepen our understanding of the role of sHsp proteins in CSCs and provide novel therapeutic strategies targeting sHsps in cancer treatment.

## 4. Small Heat Shock Proteins in Cancer Therapy

### 4.1. Anticancer Drugs Targeting sHSPs

As a majority of clinical and preclinical findings indicate sHSPs as a promising therapeutic target in cancer, a number of drugs or inhibitors have been reported and utilized to interrogate sHSPs’ roles in cancer. The reports are mainly focused on HspB1 as a molecular target for cancer therapy. Hence, in the present subsection, the drugs targeting HspB1 are analyzed in more detail. Although several drugs or compounds for other sHSPs in cancer therapy have been described, other sHSPs are omitted here because no selective inhibitors targeting these sHSPs have been reported.

Here, we summarize the drugs aimed at reducing HspB1 expression or inhibiting its actions in cancer therapy (Table 3). Small molecule inhibitors (RP101, quercetin, J2, ovatodiolide, and methyl antcinate) bind to the HspB1 protein and inhibit its function. Another strategy utilizes peptide aptamers (PA11, PA50) that bind directly to the protein and inhibit its oligomerization or dimerization. Moreover, the third approach is antisense oligonucleotide (OGX-427), which targets HspB1 mRNA and prevents translation of the protein.

Several small molecule inhibitors targeting HspB1 are currently under development: RP101, quercetin, J2, ovatodiolide, and methyl antcinate. RP101 (also known as bromovinyldeoxyuridine, BVDU, or brivudine) is a nucleoside that inhibits HspB1 function via binding with Phe29 and Phe33 of HspB1. RP101 functions as a chemo-sensitizing agent that inhibits the resistance and potentiates the effects of many chemotherapeutic drugs including mitomycin [148,149], gemcitabine [149,150], cisplatin [149,150], and cyclophosphamide [149]. In clinical studies, RP101 [149,150] or RP101 with gemcitabine [150,151] increased the overall survival rate of pancreatic cancer patients. However, overdosing of RP101 caused increased toxic side effects of gemcitabine in some patients [149], and new second-generation candidates of RP101 have been identified and are being developed for further evaluation [152]. Quercetin is a plant-derived bioflavonoid with anticancer properties [153]. It suppresses the HSF1-dependent induction of the Hsps and shows antitumor effects in gastric, oral, lymphomas, prostate, colorectal, breast, pancreatic, liver, and lung cancer cell lines and various cancer stem cells [154,155,156,157]. Quercetin can act as a chemo-sensitizer, and it enhances the antitumor effects of first-line chemotherapeutic drugs such as doxorubicin, gemcitabine, 5-fluorouracil, and cisplatin [158,159]. Interestingly, besides its inhibitory effect on HspB1 expression, quercetin can suppress HspB1 activity by impairing its phosphorylation in CSCs [138]. Despite studies showing that quercetin can be a suitable agent for cancer treatment, there are no ongoing anticancer trials for quercetin. J2, a synthetic chromone compound, can induce the crosslinking of HspB1 protein and form HspB1 abnormal dimerization, thereby inhibiting its functions [160]. Recently, ovatodiolide [132] and methyl antcinate [161], two plant-derived compounds, have been reported to decrease HspB1 protein expression in breast CSCs and inhibit CSCs. It has to be elucidated whether these compounds are clinically applicable against breast cancer.

The second approach to targeting HspB1 is the use of specific peptides, which are called peptide aptamers, to bind the protein and inhibit the functions of HspB1. Peptide aptamers are short peptides that are designed to bind to specific protein domains and disrupt the protein function. Recent research showed that two peptide aptamers, PA11 and PA50, can specifically bind to HspB1, inhibiting HspB1 dimerization or oligomerization, thereby negatively modulating the functions of HspB1 [162]. These peptide aptamers are reported to show antitumor effects in vitro [162] and in vivo [163]. Similar to the small molecule inhibitors of HspB1, a peptide aptamer is always more effective when used with other anticancer drugs than when used alone. More efforts are needed to promote the preclinical and clinical application of peptide aptamers and provide a potential application to cancer therapy.

The third approach utilizes antisense oligonucleotide (ASO) targeting HspB1 mRNA, and OGX-427, which prevents the expression of HspB1 protein. OGX-427 reduced xenograft tumor growth when used in combination with chloroquine [164] or gemcitabine, respectively [165], compared to treatment with the drug alone. The Phase I study of dose-escalation OGX-427 in prostate, bladder, breast, and lung cancers showed that OGX-427 was well tolerated at a high dose (1000 mg), and it can decrease tumor marker expression and the number of circulating tumor cells (CTCs) in patients with prostate and ovarian cancers [166]. In a Phase II trial for castrate-resistant prostate cancer (CRPC), 71% of patients treated with OGX-427 and prednisone were progression-free at 12 weeks, compared to 40% of patients treated with prednisone alone [167]. However, in another Phase II trial for metastatic non-small-cell lung cancer (NSCLC), the addition of OGX-427 to the carboplatin–pemetrexed regimen did not improve outcomes and the efficacy of first-line chemotherapy for patients [168]. More clinical studies are needed to evaluate the efficacy and side effects of OGX-427 as a combinational clinical therapy in the treatment of different cancer patients.

### 4.2. sHSPs-Based Cancer Therapy

Other than a molecular target for cancer therapy, sHsps have been reported to be used as a multifunctional scaffold for the targeted therapeutic and imaging systems in cancers. The naturally occurring small heat shock protein 16.5, which originates from *Methanocaldococcus jannaschii*, is reported to form a cage-like structure to act as multifunctional biomaterials. The genetically and chemically modified Hsp16.5 cages, Cy5.5-HspDEVD-BHQ3, were developed for imaging caspase activity in vitro and in vivo [169]. Thus, these sHsp cages may provide efficient imaging agent carriers to monitor the therapeutic evaluation by imaging caspase activity within tumors. Similarly, Hsp16.5-based nanocages, conjugated with gadolinium (III)-chelated agents and iRGD peptides, were developed for the diagnosis of pancreatic cancers by magnetic resonance imaging (MRI) [170]. It showed that sHsps have great potential in the diagnosis of cancers as a carrier to construct a specific and sensitive MRI contrast agent. Moreover, Hsp16.5-based cages carrying doxorubicin (an anticancer agent) were tested in various cancer cell lines [171] and could provide a useful drug delivery system in cancer therapy. As sHsps cages have good biocompatibility, biodegradability, and easy fabrication, they may be promising as biomedical materials for drug or imaging agent delivery in cancer therapy and other biomedical applications.

## 5. Conclusions and Future Perspectives

In summary, sHSPs are crucial to the regulation of the tumorigenesis, development, metastasis, and chemoresistance of cancers, as well as cancer stem cells, and they may act as an oncogene or a tumor suppressor in a context-dependent manner in cancer progression. In addition, biological and clinical data strengthen the idea that sHSPs are closely associated with the prognosis and progression of cancers. Moreover, sHSP-targeted drugs or inhibitors and therapeutic and imaging strategies in cancers have been developed and show great potential in cancer treatment. Considerable progress has been made in recent years in understanding the functions and mechanisms of sHSPs in cancers. However, many central aspects are still in need of further clarification. First, in various types of cancers, the precise roles of sHSPs and the detailed underlying mechanisms in cancer progression should be investigated. How do sHSPs regulate cancer progression in a context-dependent manner? Second, the precise mechanisms of the regulation of sHSP expression and activity in cancer progression need to be determined. Third, the cellular substrate molecules for different sHSPs and the dynamic interaction between sHSPs and these molecules in different types of cancers remain to be elucidated more in detail. Moreover, the discovery of therapeutic inhibitors or drugs targeted to different sHSPs has so far remained largely unexplored. As the structural complexity of Hsp27 and its dynamic interactions with substracts challenge the discovery of therapeutic inhibitors and drugs, potent and selective inhibitors of Hsp27 are still missing. In addition, most inhibitors and drugs have been developed for Hsp27; however, drugs targeting other sHsps remain largely open issues. Furthermore, recent studies have suggested combinations of sHSP inhibitors and other chemotherapy agents, and these combinations are promising in drug-resistant cancer treatment. Researchers should pay more attention to this direction when investigating potential new cancer therapies.

## Figures and Tables

**Figure 1 ijms-21-06611-f001:**
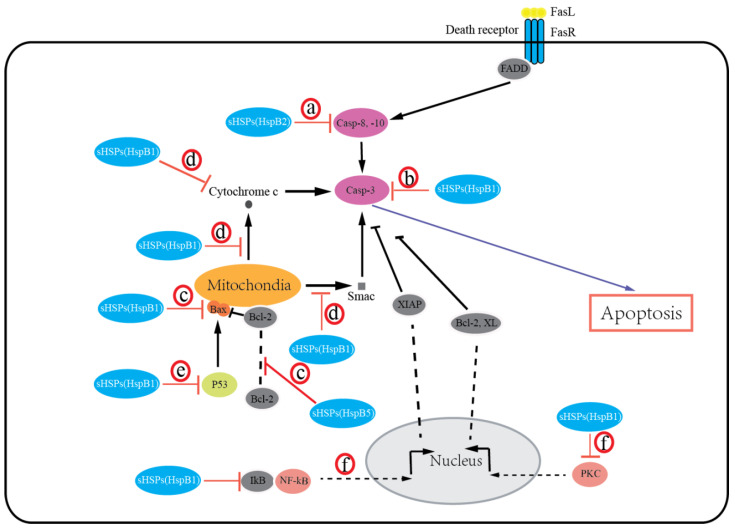
Small heat shock proteins (sHSPs) regulate apoptosis in the extrinsic and intrinsic apoptotic pathways. sHSPs inhibit activation of caspase-8, -10 (**a**) or caspase-3 (**b**) to block apoptosis. sHSPs inhibit activation of the intrinsic apoptotic pathway by interacting with bax, bak, or other members of the Bcl-2 family (**c**), or by inhibiting cytochrome c and Smac release from mitochondria, or binding with cytochrome c (**d**), or by inhibiting the p53-dependent activation of the proapoptotic Bcl-2 family members (**e**). sHSPs also interact with PKC or IkB to inhibit caspase-3 activation (**f**).

**Figure 2 ijms-21-06611-f002:**
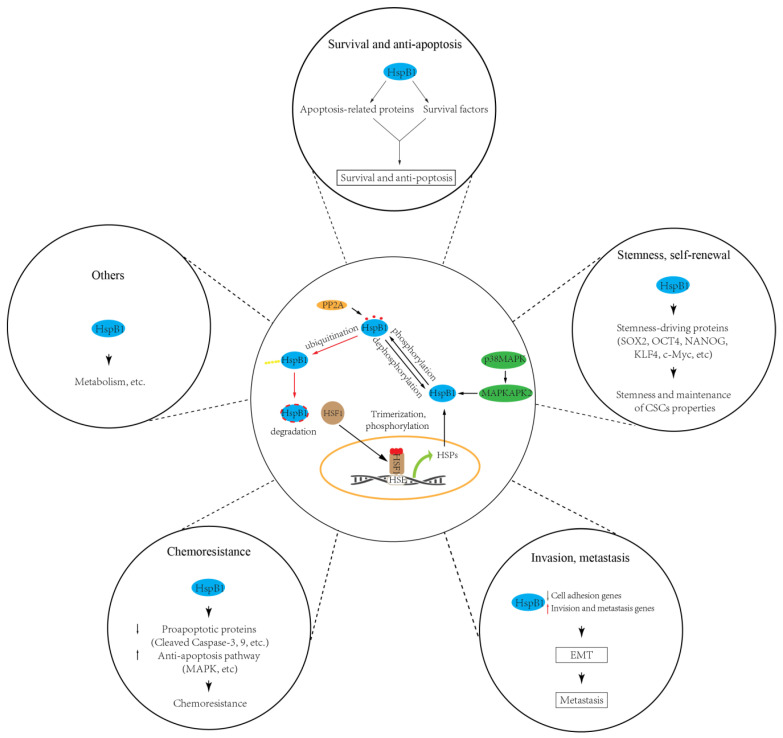
sHSPs (HspB1) regulate cancer stem cell (CSC) properties, including survival, stemness, invasiveness, chemoresistance, and others. HspB1 involvement in modulating CSC properties depends on the levels of its expression and activity. HspB1 can be upregulated by transcription factor HSF1, while its activity can be modulated by phosphorylation (e.g., by p38/MAPK pathway) and dephosphorylation (e.g., by PP2A).

**Table 1 ijms-21-06611-t001:** Human small heat shock proteins (sHSPs) and their expression ^1^.

Protein Name	Alternative Names	Molecular Mass (kDa)	Tissue Expression
HspB1	Hsp27, Hsp25, Hsp28	22.8	Ubiquitous
HspB2	MKBP	20.2	Cardiac and skeletal muscle
HspB3	Hsp17	17.0	Cardiac and skeletal muscle
HspB4	αA-crystallin	19.9	Eye lens
HspB5	αB-crystallin	20.2	Ubiquitous
HspB6	Hsp20, p20	17.1	Ubiquitous
HspB7	cvHsp	18.6	Cardiac and skeletal muscle
HspB8	Hsp22	21.6	Ubiquitous
HspB9	CT51	17.5	Testis
HspB10	ODF1	28.4	Testis

^1^ Table is based on Kampinga et al. [16] and Mymrikov et al. [17].

**Table 2 ijms-21-06611-t002:** sHSP expression in cancers.

sHSPs (Alias)	Type of Cancers	Expression Status	Functions	Reference
HspB1 (Hsp27)	Breast cancer	Overexpression	Oncogenic	[25,26]
	Cervical cancer	Overexpression		[31]
	Colorectal cancer	Overexpression		[32]
	Gastric cancer	Overexpression		[36]
	Glioblastoma	Overexpression		[55]
	Larynx cancer	Overexpression		[39]
	Lung cancer	Overexpression		[43]
	Prostate cancer	Overexpression		[51]
	Renal cancer	Overexpression		[52]
HspB2 (MKBP)	Breast cancer	Overexpression	Oncogenic	[27]
HspB3 (Hsp17)	No data	No data	No data	No data
HspB4 (αA-crystallin)	Pancreatic cancer	Underexpression	Tumor suppressive	[49,50]
HspB5 (αB-crystallin)	Breast cancer	Overexpression	Oncogenic	[28,29]
	Colorectal cancer	Overexpression		[33]
	Gastric cancer	Overexpression		[37]
	Glioblastoma	Overexpression		[56]
	Larynx cancer	Overexpression		[40]
	Liver cancer	Overexpression		[42]
	Lung cancer	Overexpression		[45]
	Oral cancer	Overexpression		[46]
	Osteosarcoma	Overexpression		[57]
	Renal cancer	Overexpression		[53]
HspB6 (Hsp20)	Colorectal cancer	Underexpression	Tumor suppressive	[34]
HspB7 (cvHsp)	Renal cancer	Underexpression	Tumor suppressive	[54]
HspB8 (Hsp22)	Gastric cancer	Overexpression	Oncogenic	[38]
	Ovarian cancer	Overexpression		[48]
HspB9 (CT51)	Testis cancer	Overexpression	Oncogenic	[35]
HspB10 (ODFP1)	No data	No data	No data	No data

**Table 3 ijms-21-06611-t003:** Summary of reported cancer drugs targeting HspB1.

Types	Names	Mechanism	Binding Sites	Reference
**Small Molecules**	RP101	Binds to HspB1 protein and inhibits HspB1 function	Phe29 and Phe33	[148,149,150,151,152]
	quercetin		No data available	[132,138,153,154,155,156,157,158,159,160]
	J2		Cysteine thiol group	[161]
	ovatodiolide		No data available	[162]
	methyl antcinate		No data available	[163]
**Peptide Aptamers**	PA11	Binds to HspB1 protein and inhibits its oligomerization or dimerization	No data available	[164,165,166]
	PA50		No data available	
**Antisence Oligonucleotide**	OGX-427	Binds to HspB1 mRNA and prevents translation of the protein	No data available	[167,168,169,170,171]

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
