# Peer review of "Small Heat Shock Proteins in Cancers: Functions and Therapeutic Potential for Cancer Therapy"

_ijms, 2020, doi:10.3390/ijms21186611_

Round 1
Reviewer 1 Report
Manuscript is in need of careful editing.
The following corrections should be made in the manuscript.
Page 1, line 39. (12-43 KDa)
Correction: (12-43 kDa)
Page 1, line 40. final structures[1,2].
Correction: final structures [1,2].
Analogous corrections should be made throughout the whole text.
Page 2, line 60, Table 1, Molecular mass (kD)
Correction: Molecular mass (kDa)
Page 3, line 86. [35] cancers
Correction: [35], cancers
Page 4, line 110. Hsp4 acts as a suppressor, and Hsp5 acts as a promoter.
Possible correction: HspB4 acts as a suppressor, and HspB5 acts as a promoter.
Page 4, line 141. cytochrome C[76],
Correction: cytochrome c [76],
Page 5, line 155. is clearly incompletely.
Possible correction: is clearly incomplete.
Page 5, Figure 1. cytochrome C
Correction: cytochrome c
Page 5, line 161. by inhibiting cytochrome C, Smac release from mitochondria, or binding with cytochrome C (d), or
Correction: cytochrome c
Page 5, line 171. Please check: head and neck squamous[90],
Page 5, line 172. Please check: inhibit cell migration and/or invasion.
Page 5, line 173. clinical data showed primary tumor expression…
Possible correction: clinical data showed that primary tumor expression…
Page 6, line 182. Although the overwhelming majority of data have indicated sHsps can…
Possible correction: Although the overwhelming majority of data have indicated that sHsps can…
Page 6, line 204. these data suggests that
Correction: these data suggest that
Page 7, line 267. was reported to dephosphorylates HspB1…
Correction: was reported to dephosphorylate HspB1
Page 8, line 283. Please check: [3.6-23]
Page 8, line 286. Please check: the interating proteins
Page 8, line 289. indicate HspB1 is responsible
Possible correction: indicate that HspB1 is responsible
Page 8, line 291. Please check: raiochemotherapy
Page 8, line 299. Please check: Third, current publications are mainly on the roles of HspB1 in CSCs.
Page 9, line 319. Please check: Another strategy utilize peptide aptamers… (utilizes?)
Page 10, line 327. Please check: that inhibit Hsp27 function… (inhibits?)
Page 11, lines 371-372. Please check: “The naturally occurring small heat shock protein 16.5, which originated from Methanocaldococcus jannaschii, are reported to form…” (is reported?)
Page 11, lines 399-400. Please check: “Moreover, the discovery of therapeutic inhibitors or drugs targeted to different sHSPs are so far remain largely unexplored.” (is unexplored?)
Page 12, line 416, Ref. 1. Protein-Protein Interactions in the Molecular Chaperone Network.
Correction: Protein-protein interactions in the molecular chaperone network.
Analogous corrections should be made in References 3, 4, 8, 10, 13, 17, 65, 66, 67, 68, 92, 100, 112, 113, 119, 125, 127, 129, 140, 143, 144, 147, 149, 150, 157.
Page 14, line 489, Ref. 29. alphaB-
Correction: AlphaB-
Page 14, line 504, Ref. 35. de Wit NJ, Verschuure P, Kappé G, et al.
Correction: de Wit N.J.; Verschuure P.; Kappé G.; King S.M.; de Jong W.W.; van Muijen G.N.P.; Boelens W.C.
Page 15, line 536, Ref. 47. in responseto chemotherapy
Correction: in response to chemotherapy
Page 16, line 592, Ref. 66. Cancers
Correction: Cancers (Basel)
Page 17, line 672, Ref. 98. Alpha-Bcrystallin
Correction: Alpha-B-crystallin
Page 20, Ref. 156. The more detailed information is needed.
Author Response
A point-by-point response to the Reviewers’ comments and suggestions (Please see the attachment for the revised manuscript)
- Page 1, line 39. (12-43 KDa)
Correction: (12-43 kDa)
Our reply:
Thanks Reviewer for pointing out this mistake. We have gone through the manuscript and corrected the mistake accordingly. (Pls see the line 25 on Page 1).
- Page 1, line 40. final structures[1,2].
Correction: final structures [1,2].
Analogous corrections should be made throughout the whole text.
Our reply:
Thanks Reviewer for pointing out these mistakes. We have gone through the manuscript and corrected all the analogous mistakes accordingly (Pls see the line 40, 41 on Page 1; the line 46, 47, 48, 49, 67, 68, 70, 76, 78, 81 on Page 2; the line 88 on Page 3; the line 117, 120, 121, 123, 134, 135, 137, 146, 147, 148, 150, 151, 152 on Page 4; the line 162, 165, 166, 173, 176 on Page 5; the line 229, 230, 232, 233, 234, 239, 241, 243, 247 on Page 6; the line 253, 254, 258, 259, 260, 264, 265, 266, 267, 269, 270, 271, 278, 279, 285, 286, 287 on Page 7; the line 310, 313, 315, 320, 322, 323, 325, 327, 329, 331, 332, 337, 340, 341, 343, 350, 351, 353, 357 on Page 8; the line 366, 368, 370, 371 on Page 9; the line 425, 426, 427, 430, 433, 436, 439, 440, 447, 448 on Page 11; the line 475, 479, 481, 484, 492, 496, 498 on Page 12).
- Page 2, line 60, Table 1, Molecular mass (kD)
Correction: Molecular mass (kDa)
Our reply:
We have corrected this mistake accordingly (Pls see the line 69 on Page 2, Table 1).
- Page 3, line 86. [35] cancers
Correction: [35], cancers
Our reply:
We have corrected this mistake accordingly (Pls see the line 98 on Page 3).
- Page 4, line 110. Hsp4 acts as a suppressor, and Hsp5 acts as a promoter.
Possible correction: HspB4 acts as a suppressor, and HspB5 acts as a promoter.
Our reply:
We have correted these mistakes as requested (Pls see the line 127 on Page 4).
- Page 4, line 141. cytochrome C[76],
Correction: cytochrome c [76],
Our reply:
We have correted this mistake as requested (Pls see the line 165 on Page 5).
- Page 5, line 155. is clearly incompletely.
Possible correction: is clearly incomplete.
Our reply:
We have correted this mistake as requested (Pls see the line 180 on Page 5).
- Page 5, Figure 1. cytochrome C
Correction: cytochrome c
Our reply:
We have corrected this mistake accordingly and Figure 1 has been replaced in the revised manuscript (Pls see the line 182 on Page 5).
- Page 5, line 161. by inhibiting cytochrome C, Smac release from mitochondria, or binding with cytochrome C (d), or
Correction: cytochrome c
Our reply:
We have corrected this mistake accordingly (Pls see the line 186 on Page 5).
- Page 5, line 171. Please check: head and neck squamous[90],
Our reply:
We have changed it to “head and neck squamous cell carcinoma [104],”. (Pls see the line 233 on Page 6).
- Page 5, line 172. Please check: inhibit cell migration and/or invasion.
Our reply:
We have changed this sentence to “HspB1 silencing in colorectal [100], prostate [101], ovarian [102], liver [103] cancers, or head and neck squamous cell carcinoma [104], HspB8 silencing in breast cancer [62], or HspB5 silencing in colorectal [105], renal [53] cancers inhibit migration and/or invasion of cancer cells”. (Pls see the line 232 on Page 6).
- Page 5, line 173. clinical data showed primary tumor expression…
Possible correction: clinical data showed that primary tumor expression…
Our reply:
We have corrected this mistake accordingly (Pls see the line 235 on Page 6).
- Page 6, line 182. Although the overwhelming majority of data have indicated sHsps can…
Possible correction: Although the overwhelming majority of data have indicated that sHsps can…
Our reply:
We have corrected this mistake accordingly (Pls see the line 245 on Page 6).
- Page 6, line 204. these data suggests that
Correction: these data suggest that
Our reply:
We have corrected this mistake accordingly (Pls see the line 271 on Page 7).
- Page 7, line 267. was reported to dephosphorylates HspB1…
Correction: was reported to dephosphorylate HspB1
Our reply:
We have corrected this mistake accordingly (Pls see the line 352 on Page 8).
- Page 8, line 283. Please check: [3.6-23]
Our reply:
Thanks Reviewer for pointing out this mistake. We have corrected this mistake by replacing reference 150 (Pls see the line 370 on Page 9).
- Page 8, line 286. Please check: the interating proteins
Our reply:
We have changed it to “HspB1 interacting proteins”. (Pls see the line 373 on Page 9).
- Page 8, line 289. indicate HspB1 is responsible
Possible correction: indicate that HspB1 is responsible
Our reply:
We have corrected this mistake accordingly (Pls see the line 376 on Page 9).
- Page 8, line 291. Please check: raiochemotherapy
Our reply:
Thanks Reviewer for pointing out this mistake. We have correted this mistake by “radiochemotherapy” (Pls see the line 378 on Page 9).
- Page 8, line 299. Please check: Third, current publications are mainly on the roles of HspB1 in CSCs.
Our reply:
Thanks Reviewer for pointing out this mistake. We have added “focused” into this sentence (Pls see the line 386 on Page 9).
- Page 9, line 319. Please check: Another strategy utilize peptide aptamers… (utilizes?)
Our reply:
We have changed it to “utilizes”. (Pls see the line 410 on Page 10).
- Page 10, line 327. Please check: that inhibit Hsp27 function… (inhibits?)
Our reply:
We have changed it to “inhibits”. (Pls see the line 423 on Page 11).
- Page 11, lines 371-372. Please check: “The naturally occurring small heat shock protein 16.5, which originated from Methanocaldococcus jannaschii, are reported to form…” (is reported?)
Our reply:
We have changed “are” to “is”. (Pls see the line 489 on Page 12).
- Page 11, lines 399-400. Please check: “Moreover, the discovery of therapeutic inhibitors or drugs targeted to different sHSPs are so far remain largely unexplored.” (is unexplored?)
Our reply:
We have changed “are” to “is”. (Pls see the line 517 on Page 12).
- Page 12, line 416, Ref. 1. Protein-Protein Interactions in the Molecular Chaperone Network.
Correction: Protein-protein interactions in the molecular chaperone network.
Analogous corrections should be made in References 3, 4, 8, 10, 13, 17, 65, 66, 67, 68, 92, 100, 112, 113, 119, 125, 127, 129, 140, 143, 144, 147, 149, 150, 157.
Our reply:
We have corrected this mistake accordingly (Pls see the line 538, 543, 545 on Page 13; the line 579, 585, 593, 603 on Page 14; the line 831, 835, 837,839 on Page 17; the line 977, 1001 on Page 19; the line 1062, 1063, 1077, 1094 on Page 20; the line 1128, 1134, 1165, 1173, 1177 on Page 21; the line 1242, 1248, 1250, 1271 on Page 22).
- Page 14, line 489, Ref. 29. alphaB-
Correction: AlphaB-
Our reply:
We have corrected this mistake accordingly (Pls see the line 729 on Page 15).
- Page 14, line 504, Ref. 35. de Wit NJ, Verschuure P, Kappé G, et al.
Correction: de Wit N.J.; Verschuure P.; Kappé G.; King S.M.; de Jong W.W.; van Muijen G.N.P.; Boelens W.C.
Our reply:
We have corrected this mistake accordingly (Pls see the line 744 on Page 15).
- Page 15, line 536, Ref. 47. in responseto chemotherapy
Correction: in response to chemotherapy
Our reply:
We have corrected this mistake accordingly (Pls see the line 780 on Page 16).
- Page 16, line 592, Ref. 66. Cancers
Correction: Cancers (Basel)
Our reply:
We have corrected this mistake accordingly (Pls see the line 836 on Page 17).
- Page 17, line 672, Ref. 98. Alpha-Bcrystallin
Correction: Alpha-B-crystallin
Our reply:
We have corrected this mistake accordingly (Pls see the line 995 on Page 19).
- Page 20, Ref. 156. The more detailed information is needed.
Our reply:
We have provided the detailed information for this reference (Pls see the line 1267 on Page 22), and updated the data from the reference in the revised manuscript (Pls see the line 481 on Page 12).
Reviewer 2 Report
Authors: Xiong J., Li Y., Tan X., Fu L.
Title: “Small heat shock proteins in cancer: Functions and…”
This review is dedicated to the important and interesting topic, namely a role of the small Hsps (sHsps) in cancer and their implication in tumor resistance to therapy. The authors summarize a large massive of the relevant data and consider a significance of the sHsps as potential targets for anticancer therapy as well as a number of inhibitors of the sHsps.
However, some important aspects have been missed by the authors:
- The very important point is an involvement of the sHsps in redox regulation and intracellular accumulation of reduced glutathione (GSH) that can affect apoptosis, chemo- and radioresistance of the malignant cell. The sHsps play one of pivotal roles in the development of antioxidant capacity of cancer cells and these sHsp-mediated antioxidative mechanisms do contribute to the tumor resistance to chemotherapy and radiotherapy. The significance of such sHsp-dependent, thiol-involving antioxidative mechanisms for cancer should be considered in the given review.
- The other important point is an ability of excess Hsp27 to partly compensate for the dysfunction of Hsp90, as the cytotoxicity of Hsp90 activity inhibitors (e.g. 17AAG) toward cancer cells can be enhanced by preventing the concomitant Hsp27 induction (see McCollum et al. Cancer Res 2006; Lee et al., Biochimie 2012, others). This relevant mechanism deserves to be mentioned and discussed in the given review.
- A large part of the manuscript is dedicated to the implication of sHsps in EMT, generation of CSCs and maintenance of the cancer stemness. In this connection, it seems strange that the authors do not cite the very relevant reference Kabakov A., et al., Cells 2020; all the more, it seems highly likely that the authors used that review article under writing their manuscript.
Without the listed (missed) points, the submitted manuscript seems somewhat incomplete, while the respective additions would improve it.
Author Response
A point-by-point response to the Reviewers’ comments and suggestions
(Please see the attachment for the revised manuscript)
- The very important point is an involvement of the sHsps in redox regulation and intracellular accumulation of reduced glutathione (GSH) that can affect apoptosis, chemo- and radioresistance of the malignant cell. The sHsps play one of pivotal roles in the development of antioxidant capacity of cancer cells and these sHsp-mediated antioxidative mechanisms do contribute to the tumor resistance to chemotherapy and radiotherapy. The significance of such sHsp-dependent, thiol-involving antioxidative mechanisms for cancer should be considered in the given review.
Our reply:
Thanks Reviewer for the kind suggestion, and we have added this point as “In addition, sHSPs could play cytoprotective roles against apoptosis induced by oxidative stress in cancer cells, and its highly expression in cancer cells results in cancer cell radio- and chemoresistance. As by-product of oxygen metabolism, reactive oxygen species (ROS) would induce oxidative stress, damage macromolecules such as proteins, lipids and nucleic acids, therefore trigger apoptotic pathways and result in cell death. Cells use the antioxidant system (e.g., antioxidant enzymes, GSH, CoQ as internally synthesised antioxidants; and vitamin E, carotenoids, etc. as externally supplied antioxidants) to neutralise free radical cellular damage and maintain normal redox homeostasis. Unlike normal cells, many cancer cells have developed efficient antioxidant systems to defend against oxidative damage caused by ROS and prevent cell death. Studies show that the expression of HspB1can significantly decrease cellular ROS level and upregulate total glutathione level [84,85]. HspB1 may act in different ways to modulate cell intracellular redox status: a. decreasing the free radical production by reducing the activities of enzymes including superoxide dismutase (SOD) [86,87], catalase [87,88], or decreasing of intracellular iron to prevent its participation in the free radical formation [89]; b. scavenging free radicals by modulating the activity of antioxidant enzymes such as glucose 6-phosphate dehydrogenase (G6PDH) [90,91], glutathione reductase (GR) [92], and glutathione peroxidase (GPx) [87,92] to increase reduced glutathione (GSH) level and downregulate ROS; c. activating transcription factors, such as nuclear factor erythroid 2-related factor 2 (NRF2), thereby activating its target antioxidative enzymes [93]. Consistent with the above-mentioned mechanisms, increased levels of sHSPs together with multiple antioxidant molecules including superoxide dismutase (MnSOD), thioredoxin reductase 2 (TXNRD2), glutathione (GSH), glutathione peroxidase (Gpx) were identified in thyroid tumor [94]. In fact, all antioxidants are working cooperatively as a complex network to maintain optimal redox balance. The roles of sHSPs in redox regulation in different cancer cells and how sHSPs coordinate the antioxidant defence network to protect against oxidative stress-induced cell death will need further investigation. Furthermore, current reports indicate that HspB1 is associated with therapeutic resistance in colon [95], liver [96], head and neck [97] cancer cells through regulation of ROS levels, thus targeting sHSPs could be a potent strategy to help overcome chemo/radio resistance in cancers.” (Pls see the line 198 on Page 6, and reference 84-97, the line 916 on Page 18).
- 2. The other important point is an ability of excess Hsp27 to partly compensate for the dysfunction of Hsp90, as the cytotoxicity of Hsp90 activity inhibitors (e.g. 17AAG) toward cancer cells can be enhanced by preventing the concomitant Hsp27 induction (see McCollum et al. Cancer Res 2006; Lee et al., Biochimie 2012, others). This relevant mechanism deserves to be mentioned and discussed in the given review.
Our reply:
Thanks Reviewer for the kind suggestion, and we have added this point as “Remarkably, it has been demonstrated that inhibition of Hsp90 is associated with the upregulation of HspB1. Hsp90 forms complexes with HSF1, and maintains HSF1 in a repressed, transcriptionally inactive form. Inhibitors or binding agents that target Hsp90 can disrupte Hsp90-HSF1 interaction, thereby free HSF-1 by dissociation from Hsp90. Activated HSF-1 is translocated into the nucleus, initiates transcription of previously silent Hsp genes, including HspB1, Hsp40, and Hsp70, leads to activation of a prosurvival process called the heat shock response, thus limits the activity of Hsp90 inhibitors and contributes to drug resistance and toxicity. Studies show that combination of down-regulation of HspB1 by gene silencing or inhibitors, with Hsp90 inhibitors could reduce the drug resistance and enhance the efficacy of Hsp90-directed therapy [126,127].” (Pls see the line 288 on Page 7, and reference 126, 127, the line 1055, 1058 on Page 20).
- A large part of the manuscript is dedicated to the implication of sHsps in EMT, generation of CSCs and maintenance of the cancer stemness. In this connection, it seems strange that the authors do not cite the very relevant reference Kabakov A., et al., Cells 2020; all the more, it seems highly likely that the authors used that review article under writing their manuscript.
Our reply:
In accordance with Reviewer’s suggestion, we added this reference in the revised manuscript (Pls see the line 379 on Page 9, and reference 151, the line 1151 on Page 21).
Reviewer 3 Report
Peer Review for “Small Heat Shock Proteins in Cancers: Function and Therapeutic Potential for Cancer Therapy”
This was a nice review covering the role that sHSPs have in cancer. With some minor changes I would recommend it to be accepted for publication.
The changes that I would recommend are as follows:
Line 24-24
You write “Small heat shock proteins are a class of the superfamily of HSPs with low molecular weight, and are ubiquitously expressed in all forms of life.” I think you should specify explicitly what the low molecular weight is (12-43kDa) here, because this is the first time the reader in being introduced to what classifies a sHSP, instead of doing in later on in lines 38-39.
Lines 48-50
You write “Under in vitro conditions, human sHSPs are often found to exist in a range of oligomeric states, although a few of them were reported to exist as monomers, dimers, or tetramers [9-12].” Can you specify which are reported to exist as monomers, dimers, or tetramers? In what state do they have to be in to be functional? This may be important because later on you talk about how drugs effect the oligomerization state of HSPB1 so it would be helpful to at least elaborate on the oligomerization state of this specific sHSP and when it is functional etc.
Lines 70-72
You write “Pathologically, sHSPs have been reported to linked to a number of diseases, such as cataracts, neurological disorders, myopathies, multiple sclerosis, aging and cancers[21-22].” This should be changed to “Pathologically, sHSPs have been reported to be linked to a number of diseases, such as cataracts, neurological disorders, myopathies, multiple sclerosis, aging and cancers[21-22].”
Lines 75-76
You write “This was the first report that Hsp27 (HspB1) expressed in meningiomas[23].” This does not make sense and needs to be reworded.
Lines 110-111
You write “The further mechanism investigation will clarify the divergent roles of sHSPs in tumorigenesis.” This is confusing and needs to be reworded to something like “Further investigation of the mechanism will clarify the divergent roles of sHSPs in tumorigenesis.”
Lines 122-124
You write “HspB8 is regulated by estrogen and can augments cancer cell progression by estrogenic stimulate[64-66], suggesting HspB8 might be a target associated with ER124 positive tumors.” This needs to be reworded to something like “HspB8 is regulated by estrogen and can augment cancer cell progression by estrogenic stimuli [64-66], suggesting HspB8 might be a target associated with ER124 positive tumors.”
Lines 149-150
You write “Recently, several researches indicated that phosphorylation of HspB1, HspB5 increases apoptosis in leukemia cell [72], breast cancer cell [75,83].” This is confusing and needs to be reworded to something like “Recently, several researchers indicated that phosphorylation of HspB1 increases apoptosis in leukemia cells [72] and phosphorylation of both HspB1 and HspB5 increase apoptosis in breast cancer cells [75,83].”
Line 151-152
You write “Given that sHSPs undergoes a number of post-translational modifications (PTMs) under physiologic and pathologic conditions (e.g., phosphorylation, acetylation and glycosylation, etc.).” This does not make sense and should be rewritten to something like “Given that, sHSPs undergo a number of post-translational modification (PTMs) under physiologic and pathologic conditions (e.g., phosphorylation, acetylation and glycosylation, etc.).”
Lines 192-195
You write “Additionally, it is noteworthy that phosphorylation may be important for sHSPs’ effects on these actions [100, 26, 39], and understanding the roles and functions of sHSPs’ PTMs on cell migration/invasion will deeper our understanding of sHSPs’ roles in malignancy.” This needs to be changed to “Additionally, it is noteworthy that phosphorylation may be important for sHSPs’ effects on these actions [100, 26, 39], and understanding the roles and functions of sHSPs’ PTMs on cell migration/invasion will deepen our understanding of sHSPs’ roles in malignancy.”
Lines 212-215
You write “Furthermore, HspB1 exhibits an alteration of its degree of phosphorylation when treatment with vinblastine, paclitaxel, doxorubicin in breast cancer cell, and its phosphorylation state is associated with chemotherapy resistance [75].” Can you elaborate more on this? Is it being phosphorylated associated with chemotherapy resistance or not being phosphorylated, or is it the degree of phosphorylation and if so what constitutes resistance and what does not? It is unclear the way you worded the sentence.
Lines 276-277
You write “Large numbers of publications have been demonstrated HspB1’s maintenance and modulation of the CSCs characteristic features are largely dependent on its interaction with client proteins.” This needs to be reworded to “Large numbers of publications have demonstrated that HspB1’s maintenance and modulation of the CSCs characteristic features are largely dependent on its interaction with client proteins.”
Lines 284-288
You write “However, the interating proteins with HspB1, and their implications and roles in formation/maintenance of the CSC phenotype remains to be elucidated.” This needs to be reworded to “However, HspB1 interacting proteins and their implications and roles in formation/maintenance of the CSC phenotype remains to be elucidated.”
Line 318-320
You write “Small molecule inhibitors (RP101, Quercetin, J2, ovatodiolide, and methyl antcinate) bind to the Hsp27 protein and inhibit its function.” It would be interesting to mention where they are speculated to bind to Hsp27 if that data is available.
- Small heat shock proteins in cancer therapy
In this section you start to refer to HspB1 as Hsp27. You should choose one or the other and be consistent throughout the text because it can be confusing to the reader.
I am confused as to why the note is at the end of the manuscript. If you are including work from certain studies it must be cited regardless of space. If you run out of space you need to make changes to the text in order to make space to cite the critical studies you are discussing. Once you do that you should delete the note.
Figures
Figure 1—The dark blue coloring for the shapes that are representative of the sHSPS makes the text hard to read. Either make the blue coloring lighter or make the text in the blue circles white.
Figure 2—The text in this figure is too small and vey hard to read. Make the text larger.
Author Response
A point-by-point response to the Reviewers’ comments and suggestions
(Please see the attachment for the revised manuscript)
- Line 24-24
You write “Small heat shock proteins are a class of the superfamily of HSPs with low molecular weight, and are ubiquitously expressed in all forms of life.” I think you should specify explicitly what the low molecular weight is (12-43kDa) here, because this is the first time the reader in being introduced to what classifies a sHSP, instead of doing in later on in lines 38-39.
Our reply:
Thanks Reviewer for the kind suggestion. We have added “(12-43 kDa)” to specify the “low molecular weight” in the revised manuscript (Pls see the line 25 on page 1).
- Lines 48-50
You write “Under in vitro conditions, human sHSPs are often found to exist in a range of oligomeric states, although a few of them were reported to exist as monomers, dimers, or tetramers [9-12].” Can you specify which are reported to exist as monomers, dimers, or tetramers? In what state do they have to be in to be functional? This may be important because later on you talk about how drugs effect the oligomerization state of HSPB1 so it would be helpful to at least elaborate on the oligomerization state of this specific sHSP and when it is functional etc.
Our reply:
Thanks Reviewer for the kind suggestion, and we have changed the above sentence to “Under in vitro conditions, human sHSPs are often found to exist in a range of oligomeric states, although a few of them such as HspB6, were reported to be stable and exist as dimers with the chaperone activity [9-10]. Recently, HspB1 was reported to exist as phosphorylated monomers, which are from a progressive dissociation of HspB1 oligomers induced by palytoxin in MCF-7 cells, and could play a protective role against palytoxin-induced cell death [11]. Similarly, another study showed that oligomer dissociation required only Ser90 phosphorylation of mammalian HspB1, while activation of thermoprotective activity required the phosphorylation of both Ser90 and Ser15 [12]. Further studies are required for understanding the role of phosphorylation of HspB1 in the interconversion of HspB1 ultrastructures between oligomeric/dimeric/monomeric forms, and the regulation of its functions.” for a better understanding (Pls see the line 49 on Page 2). We deleted “tetramers” as it should be an oligomeric form .
- Lines 70-72
You write “Pathologically, sHSPs have been reported to linked to a number of diseases, such as cataracts, neurological disorders, myopathies, multiple sclerosis, aging and cancers[21-22].” This should be changed to “Pathologically, sHSPs have been reported to be linked to a number of diseases, such as cataracts, neurological disorders, myopathies, multiple sclerosis, aging and cancers[21-22].”
Our reply:
Thanks Reviewer for pointing out this mistake. We have corrected this mistake accordingly (Pls see the line 80 on Page 2).
- Lines 75-76
You write “This was the first report that Hsp27 (HspB1) expressed in meningiomas[23].” This does not make sense and needs to be reworded.
Our reply:
We have reworded this sentence to “Hsp27 (HspB1) was initially found to be expressed in meningiomas [23]” (Pls see the line 87 on Page 3).
- Lines 110-111
You write “The further mechanism investigation will clarify the divergent roles of sHSPs in tumorigenesis.” This is confusing and needs to be reworded to something like “Further investigation of the mechanism will clarify the divergent roles of sHSPs in tumorigenesis.”
Our reply:
Thanks Reviewer for the kind suggestion, we have changed the sentence to “Further investigation of the mechanism will clarify the divergent roles of sHSPs in tumorigenesis” in the revised manuscript (Pls see the line 127 on page 4).
- Lines 122-124
You write “HspB8 is regulated by estrogen and can augments cancer cell progression by estrogenic stimulate[64-66], suggesting HspB8 might be a target associated with ER124 positive tumors.” This needs to be reworded to something like “HspB8 is regulated by estrogen and can augment cancer cell progression by estrogenic stimuli [64-66], suggesting HspB8 might be a target associated with ER124 positive tumors.”
Our reply:
Thanks Reviewer for pointing out these mistakes. We have corrected these mistakes accordingly (Pls see the line 140 on Page 4).
- Lines 149-150
You write “Recently, several researches indicated that phosphorylation of HspB1, HspB5 increases apoptosis in leukemia cell [72], breast cancer cell [75,83].” This is confusing and needs to be reworded to something like “Recently, several researchers indicated that phosphorylation of HspB1 increases apoptosis in leukemia cells [72] and phosphorylation of both HspB1 and HspB5 increase apoptosis in breast cancer cells [75,83].”
Our reply:
Thanks Reviewer for the kind suggestion, and we have changed the sentence to “Recently, several researches indicated that phosphorylation of HspB1 increases apoptosis in leukemia cells [72] and phosphorylation of both HspB1 and HspB5 increase apoptosis in breast cancer cells [75,83]” (Pls see the line 174 on Page 5).
- Line 151-152
You write “Given that sHSPs undergoes a number of post-translational modifications (PTMs) under physiologic and pathologic conditions (e.g., phosphorylation, acetylation and glycosylation, etc.).” This does not make sense and should be rewritten to something like “Given that, sHSPs undergo a number of post-translational modification (PTMs) under physiologic and pathologic conditions (e.g., phosphorylation, acetylation and glycosylation, etc.).”
Our reply:
Thanks Reviewer for pointing out these mistakes, and we have changed the sentence to “Given that, sHSPs undergo a number of post-translational modifications (PTMs) under physiologic and pathologic conditions (e.g., phosphorylation, acetylation and glycosylation, etc.)” (Pls see the line 176 on Page 5).
- Lines 212-215
You write “Furthermore, HspB1 exhibits an alteration of its degree of phosphorylation when treatment with vinblastine, paclitaxel, doxorubicin in breast cancer cell, and its phosphorylation state is associated with chemotherapy resistance [75].” Can you elaborate more on this? Is it being phosphorylated associated with chemotherapy resistance or not being phosphorylated, or is it the degree of phosphorylation and if so what constitutes resistance and what does not? It is unclear the way you worded the sentence.
Our reply:
Thanks Reviewer for pointing out this ambiguous statement, and we have changed this sentence to “serine 59 phosphorylation of HspB1 induces apoptosis of vinblastine-treated breast cancer cells [75], suggesting that inducing specifically the phosphorylation of HspB1 can improve therapeutic outcomes by circumventing drug resistance of breast cancer” (Pls see the line 281 on Page 7).
- Lines 276-277
You write “Large numbers of publications have been demonstrated HspB1’s maintenance and modulation of the CSCs characteristic features are largely dependent on its interaction with client proteins.” This needs to be reworded to “Large numbers of publications have demonstrated that HspB1’s maintenance and modulation of the CSCs characteristic features are largely dependent on its interaction with client proteins.”
Our reply:
Thanks Reviewer for pointing out this mistake. We have corrected this mistake accordingly (Pls see the line 363 on Page 9).
- Lines 284-288
You write “However, the interating proteins with HspB1, and their implications and roles in formation/maintenance of the CSC phenotype remains to be elucidated.” This needs to be reworded to “However, HspB1 interacting proteins and their implications and roles in formation/maintenance of the CSC phenotype remains to be elucidated.”
Our reply:
We have corrected this mistake accordingly (Pls see the line 373 on Page 9).
- Line 318-320
You write “Small molecule inhibitors (RP101, Quercetin, J2, ovatodiolide, and methyl antcinate) bind to the Hsp27 protein and inhibit its function.” It would be interesting to mention where they are speculated to bind to Hsp27 if that data is available.
Our reply:
Thanks Reviewer for the kind suggestion. We added the binding sites in Table 3 in the revised manuscript (Pls see the line 419 on Page 11, Table 3).
- Small heat shock proteins in cancer therapy
In this section you start to refer to HspB1 as Hsp27. You should choose one or the other and be consistent throughout the text because it can be confusing to the reader.
Our reply:
Thanks Reviewer for the kind suggestion. We replaced all Hsp27 with HspB1 (Pls see the line 403, 404, 408, 410 on Page 10; the line 417, 419-Table 3, 421, 423, 424, 435, 436, 438, 440, 443, 444, 446, 447, 449 on Page 11; the line 473, 474 on Page 12).
- I am confused as to why the note is at the end of the manuscript. If you are including work from certain studies it must be cited regardless of space. If you run out of space you need to make changes to the text in order to make space to cite the critical studies you are discussing. Once you do that you should delete the note.
Our reply:
Thanks Reviewer for the kind suggestion. We have deleted the note accordingly (Pls see the line 530 on Page 13).
- Figures
Figure 1—The dark blue coloring for the shapes that are representative of the sHSPS makes the text hard to read. Either make the blue coloring lighter or make the text in the blue circles white.
Figure 2—The text in this figure is too small and vey hard to read. Make the text larger.
Our reply:
Thanks Reviewer for the kind suggestion.We have changed the text color into white within the shapes of Figure 1 (Pls see the line 181 on Page 5, Figure 1), and inceased the text size in Figure 2 in the revised manuscript (Pls see the line 393 on Page 10, Figure 2).
Round 2
Reviewer 1 Report
no comments